The effects of arbuscular mycorrhizal fungi and root interaction on the competition between Trifolium repens and Lolium perenne

Ren Haiyan hren@njau.edu.cn
Gao Tao
Hu Jian
Yang Gaowen yanggw@njau.edu.cn
College of Agro-Grassland Science, Nanjing Agricultural University , Nanjing , China
Bezemer T. Martijn
Electronic publication date: 2017 Dec 20
Publication date: 2017
Volume: 5
Electronic Location ID: e4183
Received 2017 Sep 4; Accepted 2017 Dec 1
Copyright: © 2017 Ren et al.
Copyright year: 2017
Copyright holder: Ren et al.
License: This is an open access article distributed under the terms of the Creative Commons Attribution License, which permits unrestricted use, distribution, reproduction and adaptation in any medium and for any purpose provided that it is properly attributed. For attribution, the original author(s), title, publication source (PeerJ) and either DOI or URL of the article must be cited.
License URL: https://creativecommons.org/licenses/by/4.0/

Keywords: Arbuscular mycorrhizal fungi, Interspecific relationship, Root interactions, Competition, Productivity, Overyielding

Funding: Fundamental Research Funds for the Central Universities KJQN201601, KYZ201672, KYZ201554 Natural Science Foundation—Youth Foundation BK20160738, BK20150665 National Natural Science Foundation of China 31501996 This project was supported by the Fundamental Research Funds for the Central Universities (KJQN201601; KYZ201672; KYZ201554), the Basic research program of Jiangsu province (Natural Science Foundation)—Youth Foundation (BK20160738; BK20150665) and the National Natural Science Foundation of China (31501996). The funders had no role in study design, data collection and analysis, decision to publish, or preparation of the manuscript.

==============================
Understanding the factors that alter competitive interactions and coexistence between plants is a key issue in ecological research. A pot experiment was conducted to test the effects of root interaction and arbuscular mycorrhizal fungi (AMF) inoculation on the interspecies competition between Trifolium repens and Lolium perenne under different proportions of mixed sowing by the combination treatment of two levels of AMF inoculation (inoculation and non-inoculation) and two levels of root interaction (root interaction and non-root interaction). Overall, the aboveground and belowground biomass of T. repens and L. perenne were not altered by AMF inoculation across planting ratios, probably because the fertile soil reduced the positive effect of AMF on plant growth. Both inter- and intraspecies root interaction significantly decreased the aboveground biomass of T. repens, but tended to increase the aboveground biomass of L. perenne across planting ratios, and thus peaked at the 4:4 polyculture. These results showed that T. repens competed poorly with L. perenne because of inter and intraspecies root interaction. Our results indicate that interspecies root interaction regulates the competitive ability of grass L. perenne and legume T. repens in mixtures and further makes great contribution for overyielding. Furthermore, AMF may not be involved in plant–plant interaction in fertile condition.

Introduction

Plurispecific plant communities are usually more productive than monoculture, because the diverse communities are expected to optimize available resources due to positive interspecific interactions and species niche complementarities (Loreau & Hector, 2001; Hooper et al., 2005). So far, modern coexistence theory and contemporary niche theory are the most popular hypotheses for illustrating intra- and interspecific interactions (Letten, Ke & Fukami, 2017). These studies mostly focused on aboveground interactions and less is known about the effect of root–root interaction among plants on plant–plant interaction.

Mixtures of legumes and non-legumes which have been widely used in mixed sowing pasture and agricultural intercropping system can achieve better performance in productivity and stability (Hebeisen et al., 1997; Sanderson et al., 2013; Li et al., 2016; Elgersma & Søegaard, in press; Wang et al., 2017). Their coexistence links to intra- and interspecific interactions (Letten, Ke & Fukami, 2017; Ulrich, Jabot & Gotelli, 2017). Legumes and non-legumes compete for light, water and soil nutrient resources and also compensate for physiological traits and nutrient distribution from each other (Schwinning & Parsons, 1996; del Río et al., 2017).

Grass–clover mixtures of Lolium (grass) and Trifolium (legume) are considered as model systems in agricultural and natural grassland ecosystems (Nyfeler et al., 2009). The persistence and stability of these two species in mixed grassland depend on fertilizer type, growth period, outer press frequency and time (e.g., mowing) (Elgersma, Nassiri & Schlepers, 1998; Nassiri & Elgersma, 1998; Elgersma, Schlepers & Nassiri, 2000). In addition, Cernoch & Houdek (1994) found that soil moisture had a strong effect on the nitrogen (N) utilization of Trifolium repens and Lolium perenne. L. perenne N absorption increased along with soil moisture and further enhanced its competitive ability, while T. repens displayed a weaker N response at a higher soil moisture. These studies were more focused on aboveground factors and water and nutrients utilization; less is done with regard to their root interactions and soil microorganism effects.

Root interactions, including biological, physical and chemical interactions, occur between plants. It was found that adding N fertilizer could enhance the competitive ability of L. perenne; while adding phosphorous (P) fertilizer could enhance the competitive ability of T. repens in their mixtures (Dennis & Woledge, 1987; Davidson & Robson, 1990). Root exudates has been shown to drive interspecific facilitation by enhancing nodulation and N2 fixation, and N fixed by legumes can be transferred to grasses (Louarn et al., 2015; Li et al., 2016; Thilakarathna et al., 2016). However, whether root physical interactions will affect grass–clover interaction is still unknown. Several studies found that L. perenne had a higher competitive ability than T. repens, resulting from its fast growing root and a higher utilization efficiency in light (Luescher & Jacquard, 1991; Kleen, Taube & Gierus, 2011). Thus, physical interactions between grass and clover may increase their competition.

Arbuscular mycorrhizal fungi (AMF) are considered as key soil microorganisms and can colonize 80% terrestrial plants (Smith & Read, 2008; Brundrett, 2009). AMF can form symbiotic associations with plants and affect interspecific interactions by enlarging nutrient absorbing area of plant root systems, and further promote plant nutrient uptake and resistance to stress (Smith & Read, 2008). AMF acquire N from N sources and transfer some of this N to their host plant (Hodge & Storer, 2015). Wagg et al. (2011) found that in mixtures of Trifolium pratense and Lolium multiflorum, AMF could reduce their differences in competitive ability by enhancing the competitive power of T. pratense with its higher productivity. However, how do AMF root and their interactions affect T. repens and L. perenne in their mixtures need to be further tested.

In this study, AMF inoculation and root physical interaction of T. repens and L. perenne among their different planting ratios were investigated. We hypothesize that (1) root physical interaction can increase competition between T. repens and L. perenne; (2) AMF inoculation can relax the effect of root physical interaction on competition. It would help us deeply understand their survival strategies and coexistence of legumes and grasses, and provide effective guide for pasture management.

Materials and Methods

Experimental design

We employed a full factorial design that consisted of combinations of five plant-competition treatments, two levels of AMF inoculation (+AMF and −AMF) and two levels of root interaction treatments (+Root and −Root). Individuals of T. repens (T) and L. perenne (L) were planted in the mixed ratios: 8:0, 6:2, 4:4, 2:6 and 0:8 (T8L0, T6L2, T4L4, T2L6 and T0L8, respectively) to simulate competition between these two species. There were 20 treatments in total with five replicates per treatment. To investigate the amount of nitrogen fixed by T. repens and potentially transferred to other plants, additional three replicates were added for T8L0, T6L2, T4L4 and T2L6 in both +AMF and −AMF treatments. We established 124 microcosms under greenhouse conditions in cylindrical containers (Fig. 1). Microcosm without root interaction was achieved by eight separated plant growth pillars in which plant individuals were planted respectively (Fig. 1A). Pillar was made of 25 μm nylon mesh, which allowed arbuscular mycorrhizal mycelia access but not roots. Each pillar contained soil mixture of 188 g. The eight plant growth pillars were evenly located close to the container wall. In microcosm with root interaction, eight plant individuals were planted together and evenly spaced into the space between the container and root restriction pillar, which contained soil mixture of 1,504 g (188 g *8). Thus, the total growth space for the roots of the eight plants was the same in microcosms without root interaction and in microcosms with root interaction. The methods only tested the root physical interactions (root interaction in the following text) as roots also interact through their exudates, which cannot be avoided to go through the mesh we used in the experiment.

Figure 1 Conceptual diagram illustrating microcosms without and with root interaction.

(A) Microcosm without root interaction containing eight plant growth pillars (13 cm in height and 4 cm in diameter). There was no root interaction among plants as each plant individual was grown in pillar made of 25 μm nylon mesh, which allow arbuscular mycorrhizal (AM) mycelia access but not roots. Each pillar contained soil mixture of 188 g and one plant individual. (B) Microcosm with root interaction containing one root restriction pillar. Plant individuals were planted in the space between the container and root restriction pillar, which contained soil mixture of 1,504 g (188 g *8) and eight plant individuals. Root restriction pillar made of 25 μm nylon mesh allowed AM mycelia access but not roots. Thus, to ensure the homogeneity, the total growth space for root in (A) was same as in (B), by calculating the total soil mixture of eight plant growth pillars in (A) and then setting the same size pillar in (B).

Soil and inoculum preparation

Each container was 15 cm in height and 19 cm in diameter and was filled with 5.88 kg sterilized (25 kGy γ-irradiation) soil mixture of field soil, sand and grass peat (5:4:1 v/v/v). Soil used in the experiment was collected from Pailou experimental station of Nanjing Agricultural University (118.78°E, 22.04°N). The characteristics of the soil were measured, which contained 5.60% soil water content, 7.70 g/kg soil organic C, 18.49 mg/kg available P, 36.24 mg/kg nitrate (NO3–), 19.61 mg/kg ammonium (NH4+), 0.83 g/kg total nitrogen and pH of 7.42.

The five AMF species used were Claroideoglomus etunicatum, Glomus tortuosum, Rhizophagus intraradices, Glomus versiforme, Glomus aggregatum. Soil inoculation with AMF strains purchased from Beijing Academy of Agriculture and Forestry, China. AMF spores contained in soil inoculation were propagated in autoclaved (121 °C for 120 min) substrate (sand/soil, 1:2) with maize for two successive propagation cycles (three months for each cycle). The soil of five AMF species were mixed as inoculants with spore density of 48/g.

AMF inoculation and plant growth conditions

In microcosms without root interaction, each plant growth pillar received 6 g of AMF inoculants for +AMF treatment and 6 g of autoclaved inoculants (121 °C for 120 min) for −AMF treatment. In microcosms with root interaction, the soil mixture in the growth space between the container and root restriction pillar was inoculated with 48 g AMF inoculum for +AMF treatment and 48 g autoclaved inoculants (121 °C for 120 min) for −AMF treatment. To compensate other lost soil microbe because of sterilization, 80 ml filtrate from fresh field soil mud excluding AMF by 25 μm mesh was applied to each microcosm. Because the root of T. repens is easily colonized by rhizobium in the field, 24 ml (OD600 = 0.90) mixed rhizobium inoculants (Rhizobium sp. WYCCWR R10051 and Rhizobium sp. WYCCWR R10062) isolated from T. repens was added to each microcosm. Rhizobium inoculants were kindly provided by Dr. Zhang Junjie in Zhengzhou University of Light Industry (Zhang et al., 2016).

We used commercial seeds of T. repens (Trade name: Haifa) and L. perenne (Trade name: Maidi). The two varieties were widely used in south of China for pasture cultivation. The seeds were sterilized with 10% H2O2 for 10 min, washed with sterilized water and germinated in wet filter paper in lighting incubator (20 °C) for two to four days. And then seedlings were transplanted into each microcosms. Three seedlings of T. repens or L. perenne were transplanted to each plant growth pillar in the microcosms without root interaction. We alternated T. repens with L. perenne in the T4L4 treatment. Three seedlings of T. repens or L. perenne were randomly located in one growth pillar and another three seedlings of T. repens or L. perenne were transplanted into the opposite pillar in T2L6 or T6L2 treatment, respectively. Seedlings of T. repens or L. perenne were spaced in the same way in microcosms with root interaction.

The seedlings were grown under glasshouse conditions at 50–70% relative humidity, a temperature regime of 20–25 °C during day and night and natural lighting. Each microcosm was watered weekly, and soil water content was corrected to 15.4% (field capacity) every two weeks. One week later, the strongest seedlings were left and the other two were removed from each pillar of the microcosms without root interaction and the corresponding location in microcosms with root interaction. We randomized the location of all microcosms every two weeks.

Harvesting and measurements

Shoots were harvested nine, 13 and 16 weeks after planting to simulate mowing/grazing and reduce possible shoot competition for light. We failed to determine nitrogen transfer among plant individuals because of technical problems. Samples in these microcosms were added to each harvest. At the time of harvest at nine and 13 weeks, shoots were mowed to 5 cm height, sorted by species, oven dried at 65 °C for 72 h and weighed. During the final harvest all plants were destructively harvested, separated into shoots and roots and sorted by species. Roots of one plant individual of T. repens or L. perenne in each microcosm were frozen at −20 °C for determining AMF root colonization. Shoots and the left roots were oven dried at 65 °C for 72 h and weighed. We estimated the shoot dry matter for each species by pooling the three harvests. We assumed that plant individual of the same species in each microcosm had the same root dry matter. Thus, the dry matter value for roots used for determining colonization was estimated by the average root biomass of the same species. Shoot N content was examined using alkaline-hydrolysable diffusion method with Kjeldahl apparatus (Kjeltec Analyzer Unit 8400; FOSS, Hillerod, Sweden) (Bremner, Smith & Tarrant, 1996). Shoot P content was measured using an inductively coupled plasma emission spectrometer (TJA IRIS Advantage/1000 Radial ICAP Spectrometer; Thermo Jarrell Ash, Franklin, MA, USA), following digesting the plant tissue with trace-metal-grade nitric and perchloric acid and diluting in 100 ml of double-distilled water.

Frozen roots were cut into approximately 1 cm fragments. The root samples were cleared in 10% (w/v) KOH at 90 °C in a water bath for 60 min, and then washed and stained with 0.05% (w/v) Trypan blue. Thirty root segments from each sample were examined microscopically to assess AMF root colonization (Trouvelot, Kough & Gianinazzi-Pearson, 1986).

Statistical analysis

Competitive interactions between T. repens and L. perenne were calculated as the relative shoot biomass per individual (RYind) by the equation: RYind = Oij/Mij, where Oij is the shoot biomass per individual of plant species i grown in mixed planting microcosm of root interaction × AMF treatment combination species j, and Mij is the mean shoot biomass per individual of plant species i grown in monoculture microcosm of root interaction × AMF treatment combination species j (de Wit, 1960; Wagg et al., 2011).

Relative yield totals (RYTs) were calculated by the equation: RYTs = RY1 + RY2, where RY1 or RY2 are shoot biomass of plant species 1 or 2 in mixture divided by the shoot biomass in the monoculture of root interaction × AMF treatment combination. We used RYTs to estimate overyielding in grass–clover mixtures. Values greater than 1 indicate overyielding that a greater biomass production was observed in mixtures than the average of the two species in monoculture (de Wit, 1960; Wagg et al., 2011).

A general linear model (GLM) was used to determine the effect of planting ratio, root interaction and their interactions on AMF root colonization in the +AMF treatment. GLM was also applied to analyze the effects of AMF inoculation, planting ratio, root interaction and their interactions on shoot and root biomass, RYind, RYT, shoot N and P content. AMF inoculation, planting ratio and root interaction were treated as the fixed factors in these analyses. The data were log-transformed prior to analysis to ensure normality and homogeneity by using Shapiro–Wilk test.

As AMF inoculation and its interaction with other factors did not significantly alter any measurements (Tables S1–S3), +AMF and −AMF were treated as replicates in the following analysis. We used GLM to test the main factor of planting ratio, root interaction and their interaction on each measurement. Tukey’s simultaneous test were used to compare the variations between +Root and −Root treatment, and the level of significance was p < 0.05. LSD multiple-range test was applied to analyses the least significance differences among planting ratio treatments. All statistical analyses were performed using SAS Version 8.0 (SAS Institute Inc., Cary, NC, USA), and all figures were made by using Sigma plot 12.0 (Systat Software Inc., San Jose, CA, USA).

Results

AMF inoculation effect

AMF inoculation and its interaction with other factors did not significantly alter any measurements in our study (all p > 0.05; Tables S1–S3). The average mycorrhizal root colonization was less than 0.5% in the −AMF treatment, while AMF successfully colonized both T. repens and L. perenne in the +AMF treatment by 39% and 21%, respectively. As Fig. S1 shown, mycorrhizal root colonization of T. repens was not affected by either root interaction, planting ratio or their interactions (p > 0.05). Mycorrhizal root colonization of L. perenne was affected by planting ratio (p < 0.001). Mycorrhizal root colonization of L. perenne increased significantly along with the increasing proportion of T. repens in mixtures.

Above- and belowground biomass of T. repens and L. perenne

The aboveground biomass of T. repens and L. perenne were all significantly affected by root interaction, planting ratio and their interactions, as well as belowground biomass (p < 0.05; Fig. 2). The exception was total aboveground biomass, which was not affected by root interaction (p > 0.05). When the planting ratios ranged from T2:L6 to T4:L4 and T6:L2, the aboveground biomass of T. repens with root interaction respectively decreased from 47%, 46% to 42% in comparison with no root interaction (Tukey’s simultaneous test, p < 0.05; Fig. 2A). The aboveground biomass of L. perenne increased by 10% and 30% at the planting ratio of T4L4 and T6L2 (Tukey’s simultaneous test, p < 0.05) and did not alter aboveground biomass in monoculture and the planting ratio of T2L6 (Tukey’s simultaneous test, p > 0.05; Fig. 2B). Root interaction significantly decreased total aboveground biomass at the planting ratio of T4L4 and T8L0 (Tukey’s simultaneous test, p < 0.05; Fig. 2C). Total aboveground biomass peaked at the planting ratio of T4L4 (Fig. 2C).

Figure 2 Effects of root interaction (R), planting ratio (Ratio) and their interactions (R × Ratio) on above- and belowground biomass of T. repens (A and D), L. perenne (B and E) and the total (E and F).

T8L0, T6L2, T4L4, T2L6, T0L8 mean planting ratio of T. repens and L. perenne: 8:0, 6:2, 4:4, 2:6, 0:8, respectively. Bar groups with # indicate significant (p < 0.05) differences between +Root and −Root treatments. ***p < 0.001; **p < 0.01; *p < 0.05; ns p > 0.05.

The belowground biomass of T. repens decreased with root interaction in the mixed planting (Tukey’s simultaneous test, p < 0.05; Fig. 2D), which was similar to the responses of aboveground biomass (Fig. 2A). In monoculture, though the presence of root interaction increased belowground biomass of T. repens by 21% (Tukey’s simultaneous test, p < 0.05; Fig. 2D), the aboveground biomass of T. repens was decreased by 12% (Tukey’s simultaneous test, p < 0.05) with root interaction, indicating that the root interaction in intraspecies decreased the aboveground biomass of T. repens. In contrast, the belowground biomass of L. perenne significantly increased with root interaction in the mixed planting (Tukey’s simultaneous test, p < 0.05; Fig. 2E). In monoculture, the presence of root interaction increased belowground biomass of L. perenne by 64% (Tukey’s simultaneous test, p < 0.05; Fig. 2E), while the aboveground biomass of L. perenne was not altered with root interaction (Fig. 2B), indicating that the intraspecific root interaction did not affect the aboveground biomass of L. perenne. Root interaction significantly increased total aboveground biomass at the planting ratio of T0L8 and T2L6 (Tukey’s simultaneous test, p < 0.05; Fig. 2F)

Relative yields

The relative yield per individual (RYind) of T. repens and L. perenne were strongly influenced by root interaction, planting ratio and their interactions (Tukey’s simultaneous test, all p < 0.05; Figs. 3A and 3B). In mixtures with L. perenne, the RYind of T. repens was depressed below its RYind in monoculture by 27% in the absence of root interaction (Fig. 3A). The presence of root interaction reduced the RYind of T. repens by 54% compared with that in monoculture (Fig. 3A), suggesting that competitive pressure by L. perenne root decrease the growth performance of T. repens.

Figure 3 Effects of root interaction (R), planting ratio (Ratio) and their interactions (R × Ratio) on relative yield per individual (RYind) of T. repens.

(A) and L. perenne (B). T8L0, T6L2, T4L4, T2L6, T0L8 mean planting ratio of T. repens and L. perenne: 8:0, 6:2, 4:4, 2:6, 0:8, respectively. Data are mean ± SE. ***p < 0.001; *p < 0.05.

In mixtures with T. repens, the RYind of L. perenne was promoted above its RYind in monoculture by 65% on average in the absence of root interaction (Fig. 3B). The presence of root interaction largely enhanced the RYind of L. perenne by 101% on average compared with that in monoculture (Fig. 3B), suggesting that L. perenne benefited from interspecies root interaction. In addition, the RYind of L. perenne increased along with the proportion of T. repens in mixture (p < 0.0001; Fig. 3B). The positive effect of root interaction on the RYind of L. perenne was also raised with the increase in the proportion of T. repens in mixture (Root interaction × Planting ratio, p < 0.0001; Fig. 3B). Planting ratio heavily affected the RYT (p < 0.0001; Fig. 4). RYT peaked at the 4:4 mixture of T. repens and L. perenne.

Figure 4 Effects of root interaction (R), planting ratio (Ratio) and their interactions (R × Ratio) on the relative yield total.

T8L0, T6L2, T4L4, T2L6, T0L8 mean planting ratio of T. repens and L. perenne: 8:0, 6:2, 4:4, 2:6, 0:8, respectively. Data are mean ± SE. ***p < 0.001; ns p > 0.05.

N and P content

Shoot N content of T. repens were not affected by root interaction, planting ratio or their interactions (Table S3). Shoot N content of L. perenne significantly decreased when its planting ratio changed from monocultures to mixtures in any ratio with T. repens (LSD, p < 0.05; Table 1). Shoot P content of both T. repens and L. perenne were not significantly altered by root interaction, planting ratio or their interactions (Table S3).

Table 1 Planting ratio effects on N and P content of T. repens and L. perenne (mean ± SE).

	T. repens	L. perenne	
N (%)	P (%)	N (%)	P (%)	
T0L8	–	–	3.28 ± 0.14A	0.56 ± 0.04	
T2L6	3.49 ± 0.10	0.70 ± 0.11	2.15 ± 0.12B	0.49 ± 0.04	
T4L4	3.55 ± 0.11	0.72 ± 0.06	2.13 ± 0.13B	0.50 ± 0.05	
T6L4	3.54 ± 0.10	0.58 ± 0.05	2.41 ± 0.13B	0.52 ± 0.03	
T8L0	3.39 ± 0.11	0.49 ± 0.06	–	–	
Notes:

Different uppercases represent significant differences among planting ratio of T. repens (T) and L. perenne (L) (LSD multiple-range tests, p < 0.05).

T8L0, T6L2, T4L4, T2L6, T0L8 mean planting ratio: 8:0, 6:2, 4:4, 2:6, 0:8, respectively.

Discussion

Root interactions exert contrasting effects on T. repens and L. perenne. In mixture, the much higher yield of L. perenne suggests that it suppresses the growth of T. repens by interspecies root interactions. Overyielding in mixture is majorly contributed by the biomass of L. perenne. Although legumes could offer extra N to grasses by using its specific fixed N (Louarn et al., 2015; Thilakarathna et al., 2016), which should enhance the competitive ability of grasses, our result indicated that there was competition between T. repens and L. perenne as a lower N% of L. perenne was observed in polyculture (Table 1). Since planting ratio strongly affected the RYT, it is very crucial to select reasonable scale for optimizing mixed sowing pasture and agricultural intercropping system.

We failed to detect significant effects of AMF inoculation on shoot or root biomass of both species. This is consistent with the result from Endlweber & Scheu (2007)’s study. However, AMF inoculation reduced the competitive inequality between T. pratense and L. multiflorum by reducing the growth suppression of the legume by the grass (Wagg et al., 2011). It may attribute to the fertile soil condition. AMF inoculation has been shown to result in greater effects on the plant growth response in the less productive soil and can suppress plant growth in high P soil (Johnson, 2010; Wagg et al., 2011; Johnson et al., 2015).

No matter with or without root interaction, the mixtures of T. repens and L. perenne had no significant difference with the monocultures in regard to AMF colonization root of T. repens. However, L. perenne showed a higher AMF colonization in mixtures than in its monoculture, and increased mycorrhizal root colonization significantly along with the increased proportion of T. repens in mixtures, which is similar to competitive ability of L. perenne across planting ratios (Fig. 3B). These indicate that legume and grass mixed sowing grassland could enhance grasses’ AMF colonization and it was related to plant competition.

Extraradical hyphae of AMF affect interspecific competition with root interaction in mixtures by mediating resource competition (Janouskova et al., 2011; Wagg et al., 2011; Jiang et al., 2017). When comparing the legume T. repens and the grass L. perenne, our results showed that T. repens had twofold higher AMF colonization than L. perenne, which may result in a higher N benefit from AMF for legume species than grasses. As AMF contributes to plant N variation widely (Hodge & Storer, 2015), plant N content of L. perenne significantly decreased from monocultures to mixtures in any ratio with T. repens suggesting that this extra N conferred a competitive advantage to T. repens, and it is also likely that competition for N between symbionts occurs. Legume T. repens presented a higher N content because of its widely known N fixing capacity, resulting from its N fixing rhizobial symbionts. Besides, with root interaction, the decreased belowground biomass of T. repens in mixtures in comparison with it increased in monocultures suggests that legume could better promote their nutrient absorbing by intraspecies root interaction rather than interspecies root interaction. Our findings showed that the total yield of mixtures decreased aboveground biomass majorly because of T. repens, and increased belowground biomass majorly because of L. perenne across planting ratios. Overyielding in mixtures was mainly due to the biomass of L. perenne. It indicates that L. perenne has much stronger root interaction than legume T. repens by producing more biomass. The previous study from Mouat et al. (1987) also confirms that the root competitive capacity of T. repens is restricted by the concentration and types of soil nutrients elements. Since plant species absorbing soil nutrient largely rely on their root system, the different root interaction of T. repens and L. perenne thus affect their variable aboveground biomass. Besides, the mechanism of overcompensatory growth could also verify their root competitive capacity. L. perenne performs overcompensatory growth by decreasing root biomass or its shoot: root ratio, but T. repens is dependent on enhancing photosynthetic efficiency or symplastic growth of the shoot and root together (Faurie, Soussana & Sinoquet, 1996; Akmal & Janssens, 2004). The mixed sowing of T. repens and L. perenne decreased the intraspecific root interaction of L. perenne and increased the total productivity in mixtures, and thus contributes to overyielding effect. It is worth mentioning that the competitive capacity of T. repens and L. perenne change along with their growth periods (Schenk et al., 1995; Coutts & Jones, 2002). Although mixed sowing increase the yield of L. perenne and decrease of the one from T. repens, their competitive capabilities change in an inverse way and thus further enhance their compatibility (Elgersma, Schlepers & Nassiri, 2000; Nie et al., 2004). Therefore, they could sustainably coexist in the long term.

Building productive and stable artificial grassland is a very important measure for meeting livestock—feeds balance (Zhou et al., 2006; Huang et al., 2017). Mixed sowing of legume T. repens and grass L. perenne in half could effectively improve grassland production and forage quality. Our results suggest that planting ratios is a key driver in modulating plant species competitive power in mixtures through root interaction. Our findings provide better understanding of the root interaction and nutrient usage capabilities of these two most important forages, and have important implications for optimizing grassland measures according to the characteristic of component species.

Supplemental Information

Supplemental Information 1 Effects of root interaction (R), planting ratio (Ratio) and their interactions (R × Ratio) on mycorrhizal root colonization of T. repens (T) (a) and L. perenne (L) (b).

T8L0, T6L2, T4L4, T2L6, T0L8 mean planting ratio: 8:0, 6:2, 4:4, 2:6, 0:8, respectively. Bar groups with different letters indicate significant differences among planting ratios (n = 10 or 13). Data are means ± SE. **P < 0.01; ns P > 0.05.

Click here for additional data file.

Supplemental Information 2 F ratios and P values resulting from GLM analysis of the effects of AMF inoculation (AMF) root interaction (R), planting ratio (Ratio) and their interactions on the shoot and root biomass of T. repens and L. perenne.

Notes: Significant effects of treatments are indicated in bold.

Click here for additional data file.

Supplemental Information 3 F ratios resulting from GLM analysis of the effects of AMF inoculation (AMF) root interaction (R), planting ratio (Ratio) and their interactions on the relative yield per individual (RYind) of T. repens and L. perenne and the relative yield total (RYT) in mixtures.

Significant effects of treatments are indicated in bold. *p < 0.05; ***p < 0.0001.

Click here for additional data file.

Supplemental Information 4 F ratios resulting from GLM analysis of the effects of AMF inoculation (AMF) root interaction (R), planting ratio (Ratio) and their interactions on the shoot N and P content of T. repens and L. perenne.

Significant effects of treatments are indicated in bold. **p < 0.01; *** p < 0.0001.

Click here for additional data file.

Supplemental Information 5 Raw data.

Click here for additional data file.

We are grateful to Hong Shen, Weiyang Gui, Bin Liu, Xiao Sun, Jihui Chen, Mohan Liu who helped in collecting and processing data. Many thanks are expressed to the anonymous reviewers for their helpful suggestions.

Additional Information and Declarations

Competing Interests

Author Contributions

Data Availability

The authors declare that they have no competing interests.

Haiyan Ren conceived and designed the experiments, analyzed the data, contributed reagents/materials/analysis tools, wrote the paper, prepared figures and/or tables, reviewed drafts of the paper.

Tao Gao performed the experiments, reviewed drafts of the paper.

Jian Hu performed the experiments, contributed reagents/materials/analysis tools, reviewed drafts of the paper.

Gaowen Yang conceived and designed the experiments, analyzed the data, wrote the paper, prepared figures and/or tables, reviewed drafts of the paper.

The following information was supplied regarding data availability:

The raw data has been provided as Supplemental Dataset Files.

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
