# Peer review of "The effects of arbuscular mycorrhizal fungi and root interaction on the competition between Trifolium repens and Lolium perenne"

_PeerJ, doi:10.7717/peerj.4183_

## Round 0.1 · original submission · Major Revisions

· Academic Editor

Major Revisions

Dear authors,

Your manuscript has now been reviewed by three experts. As you will see when reading the reviews, all three reviewers see merits in your study, but they have severe comments on the writing / English style, the experimental design, the way the results are presented (or not presented), the statistical analysis and the lack of clear hypotheses. I will not repeat all the comments the reviewers made, but I have to say that based on these comments I have doubted between "rejection" and "major revision". However, I decided to offer you the chance to submit a major revised manuscript. However, this manuscript will be reviewed again, and a revision will not guarantee that the manuscript will be accepted eventually. Please take considerable attention that the statistical analyses are carried out correctly (see also the suggestions by one of the reviewers about the three way anova), that the results are properly presented and that the English / grammar is good,

Reviewer 1 ·

Basic reporting

a. Statistical results are reported in insufficient detail. Each test corresponding to the results reported in the figures needs to be reported in a table with F-values and exact p-values. Results of multiple comparisons should be reported by putting letters above bars in the figures to identify significant differences among treatment combinations.

b. The objectives paragraph (L61) is overly brief. Articulating a clearer set of objectives will assist the authors in restructuring the results and discussion sections. The objectives as written imply practical goals related to mixture development which do not reappear in any detail in the discussion.

c. The results section is is very hard to follow, and overly focussed on minor points relative to the main effects in your study. Please revise to more clearly focus on the main issues that you want the reader to see in your figures and tables. A substantial reduction in length may help to focus.

d. The dataset provided is not complete, as the individual biomass of Lolium and Trifolium in each pot does not appear to be reported, only total biomass. Also as noted below, the number of lines in the dataset (124) does not match the reported experimental design (100 pots).

e. what the error bars on the figures represent (i.e. std. dev, 1 std error) needs to be reported. No error bars are provided on figure 5. If the estimated/actual productivity values were calculated on a pot by pot basis (they should be I think), then error bars and statistical testing can be done.

f. excessive numbers of decimal places reported in the tables.

g. While the paper is generally well written, extensive editing is needed to ensure correct English grammar and smooth reading flow.

h. minor editorial points
i. please thoroughly check the ms. for typos (e.g. missing spaces between words on L2, L30, L31)
ii. L37 not clear what “higher utilization efficiency in light” means
iii. L61 not clear what competence ability is
iv. L105 colonized by n-fixing microbes?
v. L129. More explanation needed.
vi. the reference list needs to be carefully checked for misformatting and typos.

Experimental design

a. I don’t understand the pot layout based on the description in the text and figure 1. In particular, nowhere is it stated what the “plant growth pillar” is. To me it appears that in the -roots treatments plants are confined to mesh columns with some volume of soil outside the columns, while in the +root treatment the whole soil volume is available. This design is problematic because the +root plants have more rooting volume available (the equivalent of the soil in the columns plus the volume outside the columns) than the –root plants (only the volume of the columns). This is an issue as there is lots of literature out there showing that available rooting volume can influence the outcome of these types of experiments.

b. description of the experiment (100 pots) does not match with the provided dataset (124 rows).

c. Statistical analysis is not completely described or justified. Given the design a 3-way ANOVA followed by multiple comparisons is all that should be needed. I don’t understand why the 2-way ANOVA or t-tests are mentioned.

d. L93. The reasoning for the mowing treatment is not clearly stated. I understand that
forage species would routinely be mowed/grazed, however removal of the aboveground biomass is likely to trigger changes in the plant aboveground interactions that may have belowground implications.

Validity of the findings

The most important issue for the authors to deal with is the potential confound between the +root and –root treatments in terms of available rooting volume, as this may substantially affect our interpretation of the root treatment and the root by ratio and root by amf interactions. Otherwise, the underlying data in this study appear sound; the primary issues are in reporting and presentation as noted elsewhere in this review.

Additional comments

a. Overall this study has substantial merit. Interactions between Lolium and Trifolium are very well studied, thus re-examination of the many questions that have been investigated in this system under slightly different conditions is very useful.

b. I found the discussion to be quite superficial. Given the amount of research on these two species, and the number of pairwise interaction experiments involving roots, I was expecting a much wider ranging and more nuanced discussion.

c. I am surprised to see no citation to Angela Hodge’s work which has extensively used these species in a variety of experimental settings including studies involving AMF.

·

Basic reporting

I am not a native English speaker but it seems to me that the level of English used in this paper is really not good enough for a scientific publication. There are a lot of conjugation, sentence structure and grammatical mistakes. It also seems that sometimes the wrong vocabulary is used (e.g. competence instead of competitiveness? Or maybe for regeneration competence toward herbivory? But then you will have to explain more about that). The text is thus difficult to read. Therefore the English used here needs to be reviewed and improved.

There is not enough background on the two plants used in this experiment (their symbionts? one is a grass, one is a legume what does it mean for nutrient use?) and on the reasons why it is important to study them (sustainability, mix of grass and legume generally leads to a bigger productivity than monoculture and to a higher efficiency of nutrient use…). In addition, some concepts are not well explained or introduced like the plant “competence” or like the “transgressive over yielding” for instance.

Some papers cited in the introduction and in the discussion have conclusions that do not match to what is said in this draft. Some of the papers cited are quite old, (more than 20 years old). It could be nice to have newer papers cited. It is not that the older papers are not relevant but it is more because the investigation techniques have changed a lot and also the land management (in term of fertilization for instance).

The overall structure of the article conforms to an acceptable format and contains all the standard sections. The figures are relevant to the content of the article. However the quality of the Figure 1 should be improved: there is a missing parenthesis and also there are ‘arrows’ at the end of each word that should not be there I think. The legends of the figures are not detailed enough. Significance codes should be added.
All the raw data has been made available.

Experimental design

The research presented here fits to the aims and scope of PeerJ. However even if the knowledge gap being investigated is identified (i.e. ‘…many studies have explored the mechanisms of interspecific competition between T.repens and L.perenne, their works more focused on aboveground biomass, light competition, less is done with regard to their root interactions and soil microorganism effects.’), there is no clear hypothesis stated in this paper.

The investigation seems to have been conducted rigorously and the experimental design seems nice but it is not well explained. There is missing information and it is not possible to re-create the same experiment just by reading the materials and methods which is described here.

No AMF and/or no root physical interactions are non-realistic conditions in the field/nature so one could question the relevance of such a study, but it can be interesting considering fundamental knowledge.

Validity of the findings

The results presented here on the effect of AMF inoculation, roots interaction and the ratio of the two plants in mixture on L. perenne and T. repens biomasses and the conclusions made regarding the transgressive over yielding are relevant. The data on which the conclusions are based are provided. Hypotheses should be appropriately stated before to make any conclusion.

Some of the conclusions stated in this paper are made a bit quickly and are more speculations. Some of them are also going too far compared from what the results indicate.

Additional comments

Overall, I suggest that you have a native English speaking colleague review your manuscript. Some examples where the language could be improved include lines 27, 38, 45, 99,123, –and also especially in the results and in the discussion part. The current phrasing makes comprehension really difficult!!!

Introduction:
The main issue is that there is no hypothesis stated at the end of this introduction. Then you should add some background on L. perenne and T. repens and on legume/grass mixtures in grasslands. Why mixing both of these plants is important. What is the aim of doing that.
L27: You should use 'In comparison with...' or 'compared with... instead of 'In comparing with''
L28: You should check what is said by Nyfeler et al. in the publication you cite, it is different from what you wrote. Nyfeler et al. is about positive interspecific interaction and not about competition.
L36: Del Rio et al. 2016 : You should check again what is written in this publication: they are not working on L perenne and T. repens.
L39 : Goldberg and Landa 1991: You should check again what is written in this publication: they are considering seed biomass, germination, aboveground biomass, the density of the neighborhood plants etc... and not root trait and soil nutrient concentration.
L46 & L51 : I think you should avoid using « ...and so on... » in a scientific publication.
L49 : You talk about the effect of moisture on L. perenne, it could be nice to also add the effect of moisture on T. repens.
L58: « extraradical hyphae radiating from plant roots for plant-plant interaction” It is difficult to understand what you mean, you should rephrase or explain more in detail.
L60: Wagg et al. 2011 You should check again what is written in this publication. Wagg et al. work on two other species than T. repens & L. perenne.
L63: “AMF inoculation helps some species for nitrogen absorbing by changing their AMF colonization...” It is difficult to understand what you mean, you should rephrase or explain more in detail.
L66: It would be nice to explain the reason why AMF, root and their interactions affect T.repens and L.perenne in their mixtures need to be further tested.
L67: In this last paragraph of the introduction you should add the hypotheses you have.

Materials and methods:
L76: you should explain better the factors and levels of your treatments.
L79: you should precise that your are only testing the root physical interactions as roots also interact through their exhudates and you cannot avoid them to go through the mesh you are using in the experiment.
L80: “In total, 20 treatments were randomly assigned with 5 replications” you already wrote that on L75.
L80,81: I think there is a mistake in the height and diameters of the pots. Later on (L84) they are 10 times bigger.
L82: you should precise that you only keep the root from physical competitive interaction.
L83: give more details on the seed sterilization.
L84-86: It is difficult to understand how is design the experimental set up even with the figure 1: What is the plant growth pillar for? I think you should describe better the experimental set up. What is the size of the meshbags in which are the plant fort the –root interaction? What is the overall tray size? I suggest you add this information.
L84: where is the soil from? I suggest you add this information.
L88-89: are these the characteristics of the soil itself or from the miture with loess, sand and peat? I suggest you add this information.
L92: how long were the day and night duration? I suggest you add this information.
L93: could you explain the reason why you mowed all the seedlings after 2months of growth?
L99-L102: I suggest that you give more details about the nature of the AMF inoculum.You should add the total number of spore per pot and also tell if there is an equal number of spore of each of the AMF species that you used (did you use an inoculum with all 5 strains already mixed or did you use five inocula and did you mixed them yourself?)
L107- 111: In this paragraph you explain how to measure the AMF colonization in the soil. However later in the text, (in the paragraph starting L 131, in the figure) you talk about AMF root colonization. I suggest that you change that to make the text coherent.
You should precise that you are using the dry aboveground and belowground biomasses.

Results:
You should rewrite this section adding for each result the details of the statistical analyses you did (name of the test, P-value...).
You should also indicate, each time there is a significant difference if it is increasing, or decreasing and in which proportion.
Discussion
You should rewrite this section with a better english and a better structuration of the conclusions you make on each result. I also suggest that you avoid doing speculation or if you do, you should say it.
L198: Lantinga et al. 1999, Barillot et al. 2014. You should check again what is written in these publications. It is different from what you wrote: they are not saying that the resource competition is only related to lighting absorbing rather than root interaction.
L213: you are saying that T. Repens has a better nitrogen assimilation because of root interaction; but you never talk about the symbionts of T. Repens. I suggest that you discuss more about the nitrogen fixing bacteria that are more responsible for the N content of T. Repens than the root interactions.
You also need a proper conclusion where you explain after doing this experimentwhich treatment is the best to obtain transgressive overyielding and why it is important to know that for grassland management.

Figures:
I suggest that you add a figure with the nitrogen content in the plants in the different treatments.
You should add more details in you legends: number of replicates, test realized. ..You should also explain the significance code : what does mean A and B in the table 3 , what does mean *, **, *** in the figure 2, 3 and 4.
Figure 5: there is no error bar is that normal?
Raw data:
What is the ratio on the column G?
For the root and shoot biomasses: why in total? You should add the unit of measurement.
For TN and LN: I suggest that you explain the unit. It is a percentage but a percentage of what?

Reviewer 3 ·

Basic reporting

The English language in this manuscript should be improved. Currently, it was difficult for me to discern what was meant in several places. For example, I had an especially difficult time understanding what was meant by sentences starting at lines 29, 34, 44, 75, 84 and 180.

Given these issues, it was difficult for me to determine the main purpose of the experiment. If I follow the logic correctly, the researchers are trying to determine how the presence of a mutualist (AM fungi) and belowground interactions influence intra- and inter-specific competition between a forb and a grass. While the citations lead me to believe competition between these species has been tested, the authors state that no study has tested the effects of AMF on competition.

In addition to the main text, I though the explanation for the figures could use some clarity. For example, I am not certain I understand how the root interaction treatment was established, and Figure 1 did not help me much. I have some additional comments below, but, focusing on the figure, what is the osculum? Is that a hole in the bottom of the pot? I’m not sure I understand the placement of the plant growth pillars in the two treatments. It makes sense that they are around the plants in the root exclusion treatment, but what is the rationale for the placement in the treatment where root interactions are allowed? Also, I’m not sure that the dimensions for the pillars in the two treatments are accurate because it doesn’t seem possible that they are the same size in both treatments. For Figures 3 and 4, I’m not sure that I understand the x-axis. Is root biomass denoted below the zero line? If so, negative values for biomass do not make much sense. Also, I think I was confused by Figure 4 because “total biomass” typically means the sum of above- and below-ground biomass – not the sum of biomass for both the forb and the grass.

Finally, while raw data was supplied, I could not find all of the data and I believe that the column information could be better reported. For example, I’m not sure I understand what information is presented in columns “Ratio” “Shoot in Total” “Root in Total” “TN%” and “LN%”. I think that “Shoot in Total” and “Root in Total” refer to the total above and belowground biomass for the pot, and “TN%” and “LN%” refer to the %N for the forb and the grass, respectively. If that is correct, where is the biomass data for individual plants or individual species in each pot? Also, what about the AM fungal colonization/hyphae data (more on this below)?

Experimental design

As mentioned above, it was a little difficult for me to follow the motivation and justification for the experiment. However, I believe that the experimental design is interesting, and the manipulation of AMF and belowground interactions on top of a manipulation of species relative abundance is powerful for determining how these factors influence the relative importance of intra- versus inter-specific competition.

That being said, I’m not sure that I understand how belowground (root) interactions were manipulated. How was the mesh installed to prevent root competition? What are the gaps described in line 85?

I am also uncertain of some details with the AMF manipulation. How were AM fungi measured? The procedure described in the methods describes quantification of the length of extra-radical hyphae (hyphae outside of the plant root). However, the figures and results describe percent colonization of the roots, which would involve staining the fungus in the roots and measuring what percentage of the root was colonized. Which was the case? If extra-radical hyphae were quantified, what portion of the soil was used? In either case, the efficacy of the treatment could be determined. In other words, did the AMF+ pots have greater colonization/hyphae than AMF- pots?

Finally, I’m not sure I followed the methods for the statistical analyses. It states that ANOVAs were performed, but there wasn’t any information presented on whether AMF influenced plant biomass. Also, it seems like the relevant metric to look at in Figure 3 is biomass per individual plant of each species in each pot. For instance, the fact that the forb’s biomass increases with the number of forb individuals in the pot isn’t that interesting – but the response of the individual plants tells you something about how the treatments influence relative performance.

Validity of the findings

In the introduction, the addition of an AMF treatment was indicated as one of the major advances of this study. However, the effect of AMF on plant biomass (and competition) was not assessed. I’m not sure that I understand why this was not reported.

Additional comments

I believe that the design of this experiment is sound and can yield very interesting results. For instance, I was very interested in knowing how AMF influence the balance of intra- and inter-specific competition for these two species. I also think that the experiment touches on very important and timely topics in ecology: coexistence, biodiversity-ecosystem function, community consequences of mutualisms, above-below ground interactions. A solid rewrite should be able to more clearly place this work in the context of some of these ideas, making the results more accessible to a broader audience.

---

## Round 0.2 · Minor Revisions

· Academic Editor

Minor Revisions

Dear authors,

Your manuscript has now been reassessed. The reviewers both indicate that the manuscript has been improved considerably. However, there are several issues that need to be resolved before the manuscript can be accepted. The most important one is the statistical analysis. Please carefully address the issues related to this as pointed out by reviewer 1. there are also many other useful suggestions from reviewer 2 that can be incorporated relatively easily in a revision.

Reviewer 1 ·

Basic reporting

Much improved from the prev. version. Still a few typos here and there.

minor point – please provide exact p-values in Table S1

Experimental design

The description of the design is now well described; I no longer have concerns in this area.

Validity of the findings

Some remaining concerns with the statistical analyses

As described in L167-172, the authors use a 3-way ANOVA with all interactions. This is a correct approach. They then, however, (L173 and on) remove the AMF fixed effects from the model. I do not understand why they do this as it should be possible to test the main effects/interactions involving other treatments in the complete model. Running a separate model, while not strictly incorrect, overly complicates the analysis. Further, on line 176 the authors report running T-tests to compare “variations” between root interaction treatments. Two problems here, first, I assume that the authors are actually comparing means, and second, the main effects in the 3-way ANOVA should provide the answer.

The dataset and experimental description have been updated to match total number of pots (124). I am having trouble adding up the df in table S1 to reconcile the total experimental size however. Given that there are 20 pots with no T repens in the dataset and 26 with no L perenne, shouldn’t the error df. be different between the results for the two species? It is critical that ambiguities like this be resolved for me to me able to trust the reported statistical results.

Additional comments

This is my second review. The manuscript is much improved, and most importantly the authors have clarified their experimental design to make clear that the plants were grown in equivalent per-plant root volumes. The writing is much clearer now, and more concise.

·

Basic reporting

The English level of the draft has really been improved since the first submission. It is really easier to understand the manuscript but there are still a lot of conjugation, sentence structure and grammatical mistakes, so I suggest that you check again thoroughly the English.
In addition to the language, the structure and the content of the manuscript have also been really improved. The background and the cited literature have been completed and corrected like it was asked. They are still some missing information in the Statistical paragraph and some paragraphs of the discussion should be rewritten.
The overall structure of the article conforms to an acceptable format and contains all the standard sections. The figures are relevant to the content of the article. Relevant hypotheses have been added. All the raw data has been made available exept for AMF root colonization data.

All the line numbers I am referring to during the review are from the manuscript with tracked changes.

Experimental design

This part has also been really improved. There is still some missing info:
L105: you should add the gps coordinate of the station.
L111: you should add details on the nature of the AMF strains you purchased. Was it a from a spore solution? Was it from a root-organ culture?
L124: you should add the concentration of the rhizobium inoculants: how many bacteria per μl?
L153-154: ‘Roots of one plant individual were frozen at –20°C for determining AMF root colonization’: I don’t really understand. Do you mean one plant individual per microcosm? One plant individual per species per microcosm? So, in total, how many root system per treatment did you use for estimating the mean AMF colonization? It would be nice to add more details.

In the statistical analyses part:
About your data, did you check the normality and homogeneity of variances? If so which tests did you use?
For the GLM analysis, What was the most appropriate family wise errors for your data? And then, from the GLM, how were analyzed your data?
L196: you should precise that LSD stands for least significance difference.
With which test did you analyze the data from the AMF root colonization?

Validity of the findings

The data on which the conclusions are based are provided except for the AMF root colonization. Conclusion are well stated and linked to the original research questions & limited to supporting results.
Some of the conclusions stated in this paper are still made a bit quickly and are more speculations.

Additional comments

L13 :” These results showed T. repens…” there is a missing word: These results showed that T. repens…
L15: regulates
L16: makes (or you add a ‘s’ to ‘interaction’ lines 14 and 15.)
L17: plant-plant
L17: ‘involved in’ instead of ‘associated with’
L23: ‘Plant communities with more species’…’: you could replace with: Plurispecific plant communities are usually…
L31-33: Mixtures of legumes and non-legumes which have been widely used in mixed sowing pasture and agricultural intercropping system can achieve better performance in productivity and stability….
L39-40: Grass–clover mixtures of Lolium (grass) and Trifolium (legume) are considered as model system in agricultural and natural grassland ecosystems….
L44: ‘had a strong effect…’
L45: ‘L. perenne increased its N absorbing along with increased soil moisture and further enhanced its competitive ability, while T. repens had weaker N response at higher soil moisture…’ You could replace with: L. perenne N absorption increased along with soil moisture and further enhanced its competitive ability, while T. repens displayed a weaker N response at a higher soil moisture…’
L47: ‘studies’ instead of ‘works’.
L50: missing s at ‘root interaction…’ or ‘occur’
L51: Which previous study? The one from Dennis and Woledge or the one from Davidson and Robson? I think you should just write: ‘It was found that…’ or ‘Dennis & Woledge (1987) and Davidson& Robson (1990) found that…’
L59, 283, 293, 298: ‘a higher…’
L59: ‘…fast growing root system…’
L60:’… a higher light utilization efficiency…’
L65: I would remove ‘species’
L70: can you explain? How does it enhance the competitive power of T. pretense?
L 73:’ratios’
L86: ’ To investigate the amount of nitrogen fixed by T. repens and potentially transferred to other plants…’
L92: ‘The eight…’
L93: Instead of: ‘were close to container wall and evenly located in container ‘ you could write: ‘… were evenly located close to the container wall.’
L97-98:’Thus, the total growth space for the roots of the eight plants was the same in microcosms without root interaction and in microcosms with root interaction.’
L133: ‘… into each…’
L137: ‘… into the opposite…’
L143: ‘…of the microcosms… ‘
L152: ‘all plants’’
L191: remove the apostrophe after ‘its’
L192: ‘alter’ and not ‘altered’

Results:

L203 and 207: it should be p> 0.05
L203, 207 and 208: you should add the name of the test used in parenthesis before the p-values.
L203: Unless you’ve only used 1 plant per treatment to measure the root mycorrhizal colonization, you should write ‘The average mycorrhizal root colonization…’ and if you only measured the colonization on one plant per treatment then you cannot conclude anything from that.
L205: you should indicate the mean percentage of root colonization in the +AMF treatment
L214, 215: you should add the name of the test used in parenthesis before the p-values.
L228: '...by 21%...' add the statistical test and the p-value.
L229:'...by 12%...' idem
L233: '...by 64%...' idem
L240: you should add the name of the test used in parenthesis before the p-values.
L242, 244 &247 & 248: ' add the statistical test and the p-value after the%
L251,254, 255: you should add the name of the test used in parenthesis before the p-values
L258: ' Shoot N content of L. perenne significantly decreased when its planting ratio changed from monocultures to mixtures in any ratio with T. repens': add the statistical test and the p-value
L208: ‘Mycorrhizal root colonization of L. perenne increased significantly along with the increasing proportion of T. repens in mixtures.’
L216-218: you should reformulate like: ‘ When the planting ratios ranged from T2:L6, to T4:L4 and T6:L2, the aboveground biomass of T.repens with root interaction respectively decreased from 47%, 46% and 42% in comparison with no root interaction (T-test, p < 0.05; Fig. 2a)’
L218: remove 'while'
L229:'... indicating that...'
L231: '...of L. perenne significantly increased...'
L235: replace by :'indicating that the intraspecific root interaction...'
L239:'yields'
L250:'... the RYind of L. perenne was increased with the increase in the proportion of T. repens in mixture...' , replace by : '... the RYind of L. perenne increased along with the proportion of T. repens in mixture...'
L263:'... exert contrasting effect...'
L264: remove 'of these 2 species'
L266: 'legumes'
L267 :' grasses'
L268: ' which should enhance'
L269: 'as a lower'
L271:' it is' and not 'it's'
L282: ' in regard to' or 'regarding' but not 'in regarding to'

Discussion:
L293: 'which resulted in ..' this is too affirmative
L307: 'It indicates that L. perenne has much stronger root interaction than legume T. repens', you should explain and develop more this affirmation.

L294: 'contributes'
L296: 'suggesting', 'confers'
L296: you yould reformulate the sentence: '... suggesting that this extra N conferred a competitive advantage to T. repens...'
L299: You should rephrase: '... resulting from its N fixing rhizobial symbionts...'
L301: 'suggests'
L289 -303: You should rewrite this paragraph and improve the English.
L307: instead of 'was mainly from' : 'was mainly due to '
L311: 'nutrients'
L319:'participates in' or ' contributes to' instead of 'helps'
L322:'decreases the one from T. repens...'
L324-325:' thus they could sustainably cohexist.'
L329:'suggest that', 'is a key driver'

Figures and data:
Fig S1: you should add letters in top of the bars indicating if they are significant differences or not.
In the legend of the figure you should add the number of replicates used per treatment and the test used to compare the root colonization in the different treatments.

Raw data: you should add the data on the AMF root colonization in the table

In the description/title of the figures you should add what the error bar.
In the figure 2, even if you indicate the significant differences between +Root and –Root, it is not possible to know if they are significant differences between the planting ratios. I suggest that you put different letters to indicate these differences.

---

## Round 0.3 · accepted · Accept

· Academic Editor

Accept

Dear authors,

Thank you for your revision of the manuscript. I am happy to inform you that your manuscript can now be accepted for publication in PeerJ.